


# Subgrid spatial variability of soil hydraulic functions for hydrological modelling

Phillip Kreye[1] and Günter Meon[1]

[1]University of Braunschweig, Institute of Technology, Leichtweiss-Institute for Hydraulic Engineering and Water Resources, Dept. of Hydrology, Water Resources Management and Water Protection, Beethovenstr. 51a, D-38106 Braunschweig (Germany)

*Correspondence to:* Phillip Kreye
(P.Kreye@tu-bs.de)

**Abstract.** State of the art hydrological applications require a process-based spatially distributed hydrological model. Runoff characteristics are demanded to be well reproduced by the model. Despite that, the model should be able to describe the processes at a subcatchment scale in a physically credible way. The objective of this study is to present a robust procedure to generate various sets of parameterizations of soil hydraulic functions for the description of soil heterogeneity on a subgrid scale. Relations between ROSETTA generated values of saturated hydraulic conductivity ($Ks$) and van Genuchten's parameters of soil hydraulic functions were statistically analysed. An universal function that is valid for the complete bandwidth of $Ks$ values could not be found. After concentrating on natural texture classes, strong correlations were identified for all parameters. The obtained regression results were used to parameterize sets of hydraulic functions for each soil class. The methodology presented in this study is applicable on a wide range of spatial scales and does not need input data from field studies. The developments were implemented into a hydrological modelling system and were used successfully in many practical applications and projects.

## 1 Introduction

One of the major challenges in hydrological process modelling is to minimize the discrepancy between model and data scale as described e.g. by Blöschl and Sivapalan (1995) or Hopmans et al. (2002).

State of the art hydrological applications require a process-based spatially distributed hydrological model. As first objective, runoff characteristics are demanded to be well reproduced by the model. Despite that and even for large scale applications, the model should be able to describe the processes at a subcatchment scale in a physically credible way. Following Blöschl and Sivapalan (1995), hydrological processes that are dominant at spatial scales larger than the smallest calculation unit (hydrological response unit respective elementary grid size) of the model are assumed to be described directly by the model. Small scale processes below the smallest spatial calculation unit are assumed to be described indirectly by the model, e.g. by calibration. The simulation of soil water movements and storages can be particularly sensitive with respect to many model outputs (total runoff, infiltration, groundwater recharge, actual evapotranspiration etc.). Especially the water content of the soil near the surface is a decisive factor for the runoff generation (e.g. Binley et al., 1989a; Beven, 1995; de Roo et al., 1996; Coles et al.,





1997; Bronstert and Bárdossy, 1999; Entin et al., 2000; Hasenauer et al., 2009). Further, the parameterization of field saturated hydraulic conductivities ($Ks$ values, e.g. $cm\ d^{-1}$) with proxy data is an essential factor for many physically based hydrological models (Gupta et al., 2006).

Hydrological models that rely on one "effective" (specific) parameterized set of soil hydraulic functions for each soil type may

not be able to describe subgrid variation in an adequate way. Therefore it can lead to a high calibration effort and possibly to an inadequate process description. Bronstert and Bárdossy (1999), for instance, do not recommend averaged (effective) input data. Instead they suggest to use additional stochastic components to consider small scale heterogeneities. Further, Kirchner (2006) points out that *the key question is not whether models of hydrologic systems should be physically based; instead, the question is how they should be based on physics*.

Area-wide measured data of basic soil properties or even of soil hydraulic properties are not available for most hydrological model applications at the meso- and macroscale. However, in many cases rough information about the soil (e.g. soil maps) is available on a very coarse spatial resolution (1:50000 at best). Using such rough input data does not allow direct parameterization of any subgrid variability. In addition to that, soil maps are already products of regionalised input data. Consequentially, all soil hydraulic parameters based on soil maps can be interpreted (only) as effective parameters.

In this study the subgrid spatial variability for the parameterization of soil hydraulic functions will be derived indirectly from soil map information. To achieve this, three statements are formulated and will be discussed below:

1. The spatial variability of saturated hydraulic conductivity of soils on a subgrid scale can be expressed by a lognormal distribution.

2. There are relationships between the saturated hydraulic conductivity and the parameters of soil hydraulic functions.

3. These relationships are mirrored in the parameters generated by the software ROSETTA (Schaap et al., 2001). They can be used to simulate a subgrid spatial variability in a straightforward procedure, which does not require measured samples of soil properties.

The first statement was widely acknowledged in numerous studies (e.g., Law, 1944; Baker and Bouma, 1976; Sharma et al., 1980; Lauren et al., 1988; Binley et al., 1989b; Goodrich, 1990; Vauclin et al., 1994; Mallants et al., 1997; Bosch, D. D. and West, L. T., 1998; Viera et al., 2011, and many more). The second statement was investigated in several studies as well.

However, compared to the first statement, the available studies are less clear. Carsel and Parrish (1988) used approx. 3000 measurements of soil textures and bulk densities, which were summarized into 12 major texture classes. They approximated van Genuchten (1980) parameters (VGP) $\Theta_S, \Theta_R, \alpha$ and $n$ as well as $Ks$ values utilizing the empirical regression functions of Rawls et al. (1993) to describe soil hydraulic functions. In a following step, Gaussian distributions for the VGP were approxi-

mated by using the Johnson system of transformations. This was done for every VGP independently. After the transformation, high correlations were found between VGP and $Ks$ values. In a pursuing study de Rooij et al. (2004) used approx. 140 samples from two layers of an agricultural soil to fit VGP and $Ks$ values each. Relationships between the VGP and the $Ks$ values were found by means of regression analyses. However, these relationships were considered to be too weak for using the $Ks$ values as



a direct predictor for the VGP. In a next step, they used these relationships as additional information for estimating probability distribution functions for each VGP. The assumption of $Ks$ being lognormal distributed was considered as well. In a study of Li et al. (2007), data was measured experimentally to describe 63 pF curves as well as corresponding $Ks$ values, texture information, bulk densities and fractions of organic matter. The model of van Genuchten (1980) was adapted to fit the measured data

in order to obtain pF curves. This research found high correlations between VGP, measured texture classes and bulk density as well as weak correlations between measured $Ks$ values and bulk densities. No significant correlations were found between $Ks$ and the texture of the soil. Regression analyses were not conducted for $Ks$ and VGP. However, the other regressions of Li et al. (2007) indicate that there seems to be no significant relationship. Botros et al. (2009) carried out measurements to obtain pF curves for nearly 100 sediment cores. They analysed the dependence among measured $Ks$ values and VGP, which were fitted

to the measured pairs of the soil water content ($\Theta$) and the soil pressure head ($h$). Significant correlations were found between $Ks$ and $\alpha$, $n$ and $\Theta_S$; $Ks$ and $\Theta_R$ were also correlated, but did not yield significance. All these studies have in common that any analyse is always based on measured input data of soil properties. Aside from that, rather elaborate numerical simulations were necessary in many cases. As a general note, relationships between the VGP and $Ks$ values were found in many studies. Besides the lack of measured soil samples the effort of parameterization by means of sophisticated procedures that often re-

quire Monte Carlo applications is very high even for models operating on the hill slope scale. This effort is much higher for large areas and huge time scales as it is usual in e.g. climate change hydrological modelling. Consequently, the use of effective parameter sets and powerful calibration procedures is widespread. On the other hand, some kind of calibration parameters are "always" needed in hydrological modelling. Based on this, the third (innovative) statement was formulated. Premised on profound analyses of the relationship between ROSETTA generated $Ks$ values and VGP for several texture classes, the objective

of this study is to consider the subgrid spatial variability of soil hydraulic functions for hydrological modelling by using these relationships. It is worth to mention, that the methodology presented in this study is applicable for a wide range of spatial scales and does not need measured input data from field studies.

## 2  Methodology

In this section we shortly give the required theoretical background in soil physics and statistics. Further, the creation of a

database is presented by means of the software ROSETTA. The database contains the parameters and $Ks$ values for the description of pF-curves based on the equations of van Genuchten (1980). In a next step, correlations between the $Ks$ values and the parameters of the soil hydraulic functions of the generated databases are analysed.

### 2.1  Soil hydraulic functions

Since the objective of this paper is the consideration of subgrid variability of the parameterization of soil hydraulic functions

at the meso- and macroscale, the model for the description of the soil hydraulic functions has to be determined in the first place. The use of proxy information is one of very few possibilities to parameterize soil hydraulic functions extensively for large hydrological model areas. As the software ROSETTA will be used for this application (see section 2.2), the obtained





parameters are limited to the model of van Genuchten (1980). However, this model is widely used in hydrological and soil physical disciplines for describing the relation between water content and pressure head in soils:

$$\Theta(h) = \Theta_R + (\Theta_S - \Theta_R)[1 + (\alpha |h|)^n]^{-m} \tag{1}$$

There are synonymic designations for the relationship between water content and pressure head, see Durner and Flühler (2006) for details. In this study the designation "pF-curve" is used. In Eq. 1 $\Theta(h)$ denotes the volumetric water content ($cm^3 \ cm^{-3}$), $h$ ($cm$) marks the pressure head of the soil, $\Theta_R$ and $\Theta_S$ ($cm^3 \ cm^{-3}$) are defined as the residual and saturated water contents of the soil, whereas $\alpha$ ($cm^{-1}$) $n$ ($-$) and $m$ ($-$) are shape parameters of the model. Both shape parameters have a weak physical interpretation. The inverse of $\alpha$ (and also $n$) is slightly related to the air entry pressure head (however, equation 1 has no defined air entry value). $n$ is connected to the width of the pore size distribution of the soil between $\Theta_S$ and air entry pressure head. The product $mn$ is related to the width of the pore size distribution of the soil between air entry pressure head and $\Theta_R$ (Durner and Flühler, 2006; Peters and Durner, 2006). Studies of Wösten and van Genuchten (1988) and van Genuchten and Nielsen (1985) analysed the influence of these parameters on the shape of the modelled pF-curve in detail. The parameter $m$ is in most cases approximated as $1 - \frac{1}{n}$, which reduces the flexibility of the model, but enables a closed form expression for the unsaturated hydraulic conductivity by combining Eq. 1 with the pore size model of Mualem (1976):

$$K(\Theta) = K_s S_e^l \left[ 1 - \left( 1 - S_e^{\left(m^{-1}\right)} \right)^m \right]^2 \tag{2}$$

with the effective saturation $S_e$ ($cm^3 \ cm^{-3}$) as

$$S_e = \frac{\Theta - \Theta_r}{\Theta_s - \Theta_r} \tag{3}$$

In general, the absolute values of Eq. 2 are scaled by the saturated hydraulic conductivity $Ks$ ($cm \ d^{-1}$). The parameter $l$ ($-$) can be approximated as 0.5 (Kutílek and Nielsen, 1994; Hillel, 1998). The unsaturated hydraulic conductivity ($K(\Theta)$ respectively $K(h)$) can either be formulated in dependency of the soil water content $\Theta$ as shown in Eq. 2, or of the pressure head $h$.

## 2.2 Parameters for soil hydraulic functions

One objective is to investigate for correlations between ROSETTA generated VGP and $Ks$ values. To formulate statistically significant statements, a representative population for the statistical analyses has to be considered. Therefore, a short algorithm was developed to create trios of numbers within a range of 0 to 100. These trios were randomly generated with the precondition that the sum of each trio has to be 100. The numbers of each trio are assigned to be a percentage fraction of sand, silt and clay. One million fictitious samples of possible compositions of texture fractions were obtained in this manner. All three texture fractions are characterized by the same distribution with an expected value of $33.\overline{3}$ percent sand/silt/clay. The large number of generated samples was empirically determined in order to get a representative population for the statistical analyses. The regression results was stable for populations $\geq 10^5$. The number was increased to $10^6$ to safeguard validity.



The free of charge software ROSETTA (Schaap et al., 2001) was utilized to estimate the VGP $\Theta_R$, $\Theta_S$, $\alpha$ and $n$ as well as $Ks$ values per sample. It is based on neural network analyses and was calibrated by means of a large database comprised of 2134 soil samples that consists of more than 20000 pairs of $\Theta$ and $h$ in total. For the saturated hydraulic conductivity 1306 soil samples were available. 235 samples also contained data for the unsaturated hydraulic conductivity function $K(\Theta)$

respectively $K(h)$ including more than 4000 data points (Schaap et al., 1998, 2001). The database UNSODA (Leij et al., 1996; Nemes et al., 2001) contributes significantly to these data points. Additional information about early neural network applications for parameterization of soil hydraulic functions can also be found in Schaap and Bouten (1996).

The VGP sets (including $Ks$ values), obtained with ROSETTA using the randomly generated texture compositions as input, are hereafter called "database 0". In addition to this database, gradual reductions of database 0 were carried out. These reductions

were a result of the evaluation of the regression analyses. Further reasons of the reduction are given in section 3. At total four different databases were generated (database 0 and three derivatives of database 0):

1. The complete **database 0**, which consists of the total of one million VGP sets including $Ks$ values.

2. A reduced **database 1** based on the condition that $Ks < 150 \; cm \; d^{-1}$. Approx. 95% of the parameter sets of database 0 are still included.

3. A reduced **database 2** based on the condition that $Ks < 150 \; cm \; d^{-1}$ and $\Theta_R < 9\%$. Approx. 70.5% of the parameter sets of database 0 are still included.

4. Several selected **databases $3_x$**. *Variant A*: Subdivision based on natural texture classes according to the soil map of Lower Saxony, Germany. *Variant B*: Subdivision based on soil hydraulic properties.

### 2.2.1 Generation of Databases $3_x$, variant A: classification by soil map

The final reductions to databases $3_x$ were conducted for two reasons: Firstly, it is suspected that many grain size compositions in database 0 are unrealistic (e.g. 100% clay or 50% clay + 50% sand) causing the neural network of ROSETTA to extrapolate the parameters for these compositions. This may have noisy effects on possible correlations between $Ks$ and the VGP. Secondly, the presented approach is tailored to hydrological modelling at the meso- and macroscale without employing measured data. In most cases only rough information about the soil (e.g. soil maps) is available for the model area. For that reason, the

database was further reduced to obtain natural texture classes, which can be found in many soil maps. Suitable soil maps (or similar products) are widely available around the world. We used the German soil map of Lower Saxony (Edt.: J. Boess et al., 2004), see Fig. 1. Out of this, common natural compositions of grain sizes were isolated from the datasets of database 0 in order to generate databases $3_x$ (variant A). Abbreviations of the texture classes are defined in Table 1 and were assigned according to the German soil classification system (Sponagel, 2005). A pre-defined texture class for boggy soils (Hn) is not available.

Silty clay (Tu) has similar properties as clayey loam (Lt), therefore these two texture classes (Hn, Tu) are not included in the following analyses. Instead, the texture classes for silty loam (Lu) and pure sand (Ss) were added. These texture classes are not shown in the soil map (Fig. 1). However, both are contained many times in other soil maps of Germany. Around each texture



fraction, a $\pm 5\%$ boundary in each direction was considered in order to get a representative number of van Genuchten datasets for the regression analyses. Note that at total more than $10^5$ parameter sets of database 0 are still included in the databases $3_x$ (variant A). The procedure to obtain the VGP and $Ks$ values is graphically shown in Fig 3.

##### 5   2.2.2   Generation of Databases $3_x$, variant B: classification by cluster analyses

Twarakavi et al. (2010) introduced a procedure to classify soils based on their hydraulic properties. To achieve this, they used the k-means clustering algorithm. The same algorithm was used in this study to subdivide database 0 by means of hydraulic properties. This algorithm is available in MATLAB. We normalized the VGP to avoid scale effects that influence the weightings in a negative way. Minimization of euclidean distance was applied as objective function. The number of resulting subdivisions 10   (classes) is freely adjustable. We used 255 different target clases, starting with two and going up to 5680 classes.

#### 2.3   Regression analyses for soil hydraulic parameters

A flexible exponential regression model is used, since the modalities of the relations between the $Ks$ values and the VGP are unknown:

$$f(x) = ae^{bx} + ce^{dx} \tag{4}$$

where $a, b, c$ and $d$ $(-)$ are fitting parameters and $e$ $(-)$ is Euler's number. The model is adjusted by means of the Levenberg-Marquardt algorithm (Marquardt, 1963).

In addition to the univariate regression model shown above, a multivariate regression will be performed by using a general multivariate model, which be denoted as:

$$\boldsymbol{Y_{n\mathrm{x}d}} = \boldsymbol{X_{n\mathrm{x}(p+1)}} \boldsymbol{B_{(p+1)\mathrm{x}d}} + \boldsymbol{E_{n\mathrm{x}d}} \tag{5}$$

where the matrix $\boldsymbol{Y}$ denotes the dependent variables, which are assumed to be correlated among themselves. The matrix $\boldsymbol{X}$ includes the independent variables, the matrix $\boldsymbol{B}$ comprises the fitting coefficients and $\boldsymbol{E}$ gives the matrix of residuals. The index $n$ denotes the number of samples, $d$ the number of subjects and $p$ the number of predictor variables.

To evaluate the quality of the regressions, the coefficient of determination $R^2$ is calculated as follows (Sachs, 2004):

$$R^2 = \frac{SSY - RSS}{SSY} = \frac{MSS}{SSY} = 1 - \frac{RSS}{SSY} = 1 - \frac{\sum_{i=1}^{n} (y_i - \hat{y}_i)^2}{\sum_{i=1}^{n} (y_i - \bar{y})^2} \tag{6}$$

with

$$\bar{y} = \frac{1}{k} \sum_{i=1}^{k} y_i \tag{7}$$

$SSY$ is the total, $RSS$ is the residual and $MSS$ is the regression sum of squares. By normalization of $MSS$ with $SSY$ the coefficient of determination $R^2$ is obtained. $y_i$ denotes a data value and $\bar{y}$ describes the average of all data values, whereas $\hat{y}_i$



symbolizes a computed value of the regression model. $R^2$ ranges from 0 (no relation) to 1 (perfect fit).

For consideration of non-linearities, Spearman's rank correlation coefficient $r_{spear}$ can be calculated in addition to the coefficient of determination (Sachs, 2004):

$$r_{spear} = 1 - \frac{6 \sum_{i=1}^{k} (rg(x_i) - rg(y_i))^2}{k(k-1)^2} \tag{8}$$

$rg(x_i)$ and $rg(y_i)$, which are sorted into ranks ($rg$), are the values of the dataset and the fitted model with the total number of $k$. $r_{spear}$ has a range from -1 to 1, whereby 0 denotes no correlation and 1/-1 describe a perfect positive/negative correlation, respectively. Different regression analyses were conducted based on the databases.

# 3   Results and discussion

## 3.1   Regression analyses

### 3.1.1   Complete database 0 and reduced databases 1 and 2

Regression analyses based on Eq. 4 were performed for the database 0 and for the reduced databases 1 and 2 each.

The $Ks$ values in relation to the $\Theta_R$ values resulted in low correlations with $R^2$ of 0.43. A more structured $Ks - \Theta_R$ relation seems to arise for $Ks$ values smaller than 150 $cm\ d^{-1}$ and $\Theta_R$ smaller than 9%. Consequently, database 0 was reduced to database 2 and $R^2$ of the regression function, that was computed out of the complete database 0, increased to 0.72. However,

to obtain a function on the basis of database 2 new regression analyses were conducted leading to $R^2$ of 0.74. This function shown in the first plot of Fig. 5.

A similar approach was applied to evaluate $Ks$ and $\Theta_S$; no significant correlations were obtained. Because of the high correlations found for $Ks - \Theta_R$ in database 2, the reduction of the database 0 was also applied for $\Theta_S$. However, only the range of the $Ks$ values was reduced, leading to database 1. In contrast to $Ks - \Theta_R$, no significant correlations were found between $Ks$

and $\Theta_S$ based on the reduced database, see the second plot of Fig. 5.

Low correlations ($R^2 = 0.41$) were found for the parameter $n$ when using database 0. An even lower fit ($R^2 = 0.25$) was obtained when reducing database 0 to database 1 as seen in the third plot of Fig. 5.

The analysis of $Ks$ versus $\alpha$ shows neither correlations for database 0 nor for database 1 (fourth plot of Fig. 5).

Generally, in some sections of the scatter diagrams there seem to be more connections between the $Ks$ values and parameters

of the soil hydraulic functions than in other sections. However, these connections are very low and too uncertain for hydrological modelling purposes. A reduction of database 0 to database 1 respectively database 2 had a positive effect on the regression of $\Theta_R$ only. Apparently, it is not possible to obtain four single regression functions, one for each parameter.

### 3.1.2   Databases $3_x$, variant A: classification by soil map

**Univariate regression analyses**

Regression analyses based on Eq. 4 were performed for each of the natural texture classes. Concerning $\Theta_R$, very high $R^2$





between 0.88 and 0.99 were found for 7 out of the 10 texture classes with an average $R^2$ of 0.96. The other three classes reached correlations with $R^2$ lower than 0.5; therefore, these classes were not included in following analyses and applications. Generally, curves with a $R^2$ lower than 0.5 are not illustrated in the figures and tables. The regression curves of $\Theta_R$ are exponentially decreasing proportional to decreasing $Ks$ values. We have to keep in mind that van Genuchten's $\Theta_R$ has no clear physical interpretation and other fitting models for the pF-curve actually have no residual water content (see e.g. Rossi and Nimmo (1994)). The high correlations between $\Theta_R$ and $Ks$ may have to be considered as a kind of black box correlation that is valid for the ROSETTA fed van Genuchten model only. On the other hand, it is unclear how much of the found correlations may also be an artefact of the ROSETTA neural network.

Concerning $\Theta_S$, high $R^2$ between 0.68 and 0.93 were found for 5 texture classes with an average $R^2$ of 0.82. The behaviour of these classes can be divided into two groups. Group one includes Lu and Ls, group two includes Us, Sl and Su. The main textural difference of these two groups is the fractional higher clay and lower sand content in group one compared to group two, as seen in Table 1. This has an effect on the slopes of the fitted regression models. Group one shows decreasing values of $\Theta_S$ with increasing $Ks$ values, group two behaves the other way round. Assuming higher sand fractions causing higher $Ks$ values, the grain size compositions of group one are shifted in the direction to the centre of the texture triangle. This may cause smaller values of $\Theta_S$. On the other hand, moving away from the centre of the texture triangle with higher fractions of sand (as for group two) may have the opposite effect of increasing porosity. Both effects are imaginable, however, we do not want to overinterpret the physical impact of van Genuchtens's $\Theta_S$ that is based on neural network estimates.

Concerning $\alpha$, high $R^2$ values between 0.67 to 0.96 were found for four texture classes with an average $R^2$ of 0.75. As given in section 2.1, the parameter $\alpha$ is weakly related to the inverse of the air entry suction (not to forget that van Genuchten curves have no defined air entry value). In general, without specializing on van Genuchten's model, the entry suction should be higher for fine grained as for coarse grained soils. This means that the entry suction should rather decrease with increasing $Ks$ than increase. This connection cannot be found for the texture class Lu. That's why this regression (Lu) is not considered in the subsequent analysis.

Concerning $n$, very high $R^2$ between 0.63 to 1.00 were found for 7 texture classes with an average $R^2$ of 0.85. Especially for the two sandy texture classes highly accurate fits were obtained. Under the assumption of $n$ being related to the pore size distribution, many different pore sizes lead to low values of $n$, whereas many pores with a similar size lead to high values of $n$. In general, soils that are located near the borders of the texture triangle tend to have a more narrow pore size distribution as soils located in the middle of the triangle. Taking into account that these soils (pure sand, pure silte) may have higher $Ks$ values compared to loamy soils, increasing $Ks$ may be related to increasing values of van Genuchtens $n$. Again, we have to be careful not to overstretch connections of ROSETTA generated VGP to measurable physical properties of soils.

All statistical quality values from the univariate regression analyses are listed in Table 2. Additionally, p-values are included. Low p-values indicate a correlation between $Ks$ and the parameters of the soil hydraulic functions. All p-values of Table 2 are nearly zero, yielding that all shown correlations are significant. Further, the square of $r_{spear}$ yields approximately $R^2$ for most cases. This seems to validate $R^2$ as a quality criterion for the regression analyses.





**Multivariate regression analyses**

Regression analyses based on Eq. 5 were performed for each of the natural texture classes. We used $\log(Ks)$ to fill the matrix $\boldsymbol{X}$. The matrix $\boldsymbol{Y}$ comprises $\Theta_R$, $\Theta_S$, $n$ and $\alpha$. These more elaborate procedures, which consider the correlations among the dependend variables, serve as references for the previous results.

Both the shape of the obtained fits of the multivariate method and the $R^2$ turned out to be very similar to those of the univariate method. The average $R^2$ for the univariate method equals 0.84 and the average $R^2$ of the multivariate method is 0.83. The shapes of the functions differ just slightly or are even identical. Figure 7 shows the univariate and multivariate regression results for $n$ based on the texture class Su. It can be seen that both curves behave very similar with small differences at high $Ks$ values. However, $R^2$ are equal to each other and a "better" fit cannot be pointed out. All other comparisons between the

regression results of the two methods act similar to Fig. 7.

The high accordance of both method's results speaks for the robustness of the less elaborate univariate method. Based on this, the results of the univariate regression analyses will be used for further applications.

### 3.1.3   Databases $3_x$, variant B: classification based on soil hydraulic properties

**Results of the subdivision**

Fig 2 shows subdivisions of the soil texture based on soil hydraulic properties by means of cluster analyses for a number of 31 classes. Results of Twarakavi et al. (2010) showed that the subdivisions based on soil hydraulic properties are similar to the US texture based classification, especially for coarse textured soils (sands). These similarities were not found for fine textured soils. The results of our subdivision based on soil hydraulic properties are unlike to the texture based classification. However, this is not directly a contradiction to Twarakavi et al. (2010). They used the US texture triangle for comparison and we use the

german classification. In addition to that, the rules and conditions for the algorithm of the cluster analyses have a high influence on the result.

**Univariate regression analyses**

In variant B we concentrate on univariate regression analyses only. In Fig. 4 the average $R^2$ are shown in dependency of the

number of classes used for the subdivisions. As previously, regression results with $R^2$ lower than 0.5 are not considered. The abscissa is limited to a maximum of 200 classes. If more classes are used, the average $R^2$ does not increase significantly. The average $R^2$ ranges therefore mainly between 0.7 and 0.8. If we use 31 classes, which is the same number of subdivisions as the texture based classification of the german soil classification system the average $R^2$ is 0.74 and 40% of the regression results have coeffcients of determination higher than 0.5. The maximum can be found for the number of 2128 classes ($R^2$=0.82 with

49% of the regression results with $> 0.5$). The results of the regression analyses based on databases $3_x$ (variant A) yielded in an total $R^2$ of 0.88 by using nine natural texture classes and 67% of the regression results had an $R^2 > 0.5$. In additio, the application of the univariate method is faster and less elaborative. For those reasons, we will use the results of the regression analyses based on databases $3_x$ (variant A) for further applications.





## 3.2 Applications on soil hydraulic functions

Figure 8 illustrates the impact of the regression results that were obtained by the univariate method of databases $3_x$ (variant A) on van Genuchten's soil hydraulic functions for the texture classes S, Su and Lu. These three texture classes are assigned to be representative for all classes that were investigated. In addition, a wide range of $Ks$ values is covered. $Ks$ values were

selected ranging from the minimum to the maximum values that were obtained out of database $3_x$ (variant A). The pF curves of the texture class S are shown in Fig 8a. Van Genuchten's $n$ was computed out of the regression function. The pF curve of the regression with the smallest $Ks$-value has a clearly smoother slope compared to the pF curve that was obtained for the largest $Ks$-value. The lower the $Ks$ the more moves the shape of the pF curves in the direction of typical pF curves for sandy soils with a fraction of silt. The curves for low $Ks$ values tend to have a higher usable field capacity possibly leading to higher

rates of transpiration in hydrological modelling applications. The curves for the unsaturated hydraulic conductivity $K(h)$ of the texture class S are given in Fig 8d. The same parameters as for the pF curves were used. Near saturation the curves of large $Ks$ values are above the curves of low $Ks$ values. This relation changes after an intersection point at pF of approx. 2 caused by the variation of van Genuchten's $n$ that is directly connected to the parameter $m$. From the physical point of view, the shapes of the curves can be described as reasonable. The curves with lower $Ks$ values have a higher fraction of small pores. These

fraction of small pores are able to transport water for a wider range of pF in contrast to the curve parameterizations with high $Ks$ values. This leads to the intersection point that changes the dominating impact factor on the conductivity curves: For pF $< 2$ the $Ks$ value, which simply scales the curve, is the dominating factor. For pF $> 2$ van Genuchten's $m$ is the dominating impact factor. However, after the intersection point $K(h)$ is already at very low values. Therefore, the variation of $m$ for sandy soils may have a small impact compared to the impact of variations of the $Ks$ values.

Figure 8b shows the impact of the regression results on the pF curves of the texture class Su. Similar to Fig 8a, the curves for low $Ks$ values have a smoother slope. In addition to that, the modifications of van Genuchten's $\alpha$ causes the water content dropping at higher $pF$ values for the curves of low $Ks$ values compared to the curves of high $Ks$ values. This behaviour is typical for texture classes that have a slightly larger fraction of fine pores than the "standard Su". The usable field capacity is more or less the same for all $pF$ curves. The impact on hydrological model applications might nevertheless be immense

depending on the method that reduces the potential evapotranspiration to the actual one: Methods based on the actual water content of the soil within the root zone probably calculate higher rates of actual evapotranspiration using the parametrization based on low $Ks$ values than using the ones of higher $Ks$ values. On the other hand, methods based on $pF$ values of the soil are expected to be less affected. The impacts on the conductivity curves for the texture class Su are plotted in Fig 8e. Here again, an intersection point can be located (at a pF of approx. 1.8). Above this pressure head, the curves of high $Ks$ values

drop below the curves of small $Ks$ values. In contrast to the conductivity curves of the texture class S, the values of $K(h)$ at the intersection point (and close below) are still high enough to enable a water movement that is not negligible. For that reason soil water simulations are influenced especially during dry seasons.

The pF curves for the texture class Lu are visualized in Fig 8c. Here, a shift on the ordinate can be observed, whereas the curves for low $Ks$ values induce higher water contents than the curves for high $Ks$ values for the same pressure head. This is





due to the relation that was found for Lu of $\Theta_R$ and $\Theta_S$ being inverse proportional to $Ks$. However, the variations of $n$ cause different slopes of the curves. The impact on the reduction of the potential evapotranspiration is comparable to the impact described for the texture class Su. The impact on $K(h)$ is primary driven by the variations of the $Ks$ values, as seen in Fig 8f. The intersection point is approximately at pF 4. At this high pF, $K(h)$ has dropped magnitudes below the saturated value.

5 It can be summarized that the modifications of the VGP caused by the regression results of the databases $3_x$ (variant A) lead to plausible $pF$ curves. Further, the impact on the conductivity functions near saturation is primarily driven by the value of $Ks$. As the $Ks$ value works as a scaling factor for the conductivity curves, this results is no surprise and not induced by the regression functions. For medium and low saturations however, the impact is dominated by the variations of the parameterizations of the soil hydraulic functions that were produced by the regression functions. Especially for the texture Su (and similar ones)

10 the impact of the regression functions will have an impact on long term hydrological model applications. Taking the soil map of Lower Saxony for instance, texture classes with compositions like Su, Sl or similar occupy more than one third of the total area. For many of the texture classes all four VGP could be fitted in dependency of $Ks$. However, this did not always work as seen in Table 2. Following this, the correlation matrices of the VGP, generated within the regression analyses of databases $3_x$ (variant A), were taken into account more deeply. It turned out that correlations were very low between VGP, which are

15 related to $Ks$, and VGP, which are not related to $Ks$. These findings indicate the admissibility of fitting less than four VGP in dependency of $Ks$.

## 3.3 Generating subgrid spatial variability

Spatial resolutions of hydrological models mainly depend on the resolutions of the input data of soil properties and land use

20 each. These input data are often not equally resolved in space and time (e.g. the German ATKIS database). If the model area is subdivided into polygons by the hydrological model, the spatial resolution is unequally distributed and given automatically by the input layers. If the model area is subdivided into raster cells, the spatial resolution is equally distributed and depends both on input layers as on the user's interests. For latter types of models, the spatial resolution may often induce a pseudo accuracy, because the chosen grid size can be much smaller than most of the subdivisions of the input layers. In any case, the "real" spatial

25 resolution of a hydrological model that has to be considered for the process description is given by the spatial resolutions of the input data. In most cases these spatial resolutions are rather coarse causing that many processes are not directly resolved by the model.

To consider the spatial variability of soil water processes that are not directly resolved by the hydrological model, the following procedure is elaborated in order to generate parametrizations of soil hydraulic functions:

30 1. Acquisition of a soil map for the model area (or similar information). In this study: German soil map of Lower Saxony, see Fig. 1.

2. If not already included in the soil map: Transformation of soil classifications into texture information. In this study: Usage of the German soil classification system, see Sponagel (2005).



3. Randomly generation of trios of numbers within a range of 0 to 100 with the precondition that the sum of each trio has to be 100. The numbers of each trio are assigned to be a percentage fraction of sand, silt and clay.

4. Obtaining texture classes out of the soil map. Example: Sl with 65% sand, 25% silt and 10% clay (see Table 1).

5. Consideration of a boundary in each direction (sand, silt, clay). In this study: ±5% boundary. Example: Sl with 65±5% sand, 25±5% silt and 10±5% clay.

6. Generation of VGP sets with the software ROSETTA for the obtained data.

7. Regression analyses between $Ks$ values and all other VGP for each texture class.

The total number of needed randomly generated numbers (point 3) may differ in dependency of the texture classes that are going to be analysed. The ROSETTA underlying database has more samples of sandy soils than of clayey soils (Leij et al., 1996; Nemes et al., 2001). Furthermore, some combinations in the texture triangle are very seldom in nature. To ensure that these disagreements do not bias the regression results, only a close range (± boundary) near natural occurring texture classes that are obtained from soil maps should be considered for the regression analyses (here: generation of database $3_x$ (variant A), see section 2.2). The boundary was assigned to be ±5% in order to get a representative number of VGP sets for each texture class. Other values for the boundary were tested, whereby much lower values (e.g. ±1%) lead to a very close range of the $Ks$ values. Much higher values for the boundary (e.g. ±10%) blurred the VGP sets of the texture classes (there was no difference left between certain texture classes). Therefore we recommend a value of ±5% for the boundary.

At a next step, the obtained regression functions have to be applied in a hydrological model. The following procedure is recommended:

1. Assumption of a lognormal distribution for the $Ks$ values of each texture class. The mean values are given by the $Ks$ values that were obtained with ROSETTA at the center of each texture class. The standard deviations are given by the user.

2. Calculation of percentiles out of the lognormal distribution of the $Ks$ values (e.g. 10%, 30%, 50%, 70% and 90%).

3. Calculation of variations of the other VGP by using the regression functions and the $Ks$ percentiles.

4. Run the model by parallely using the VGP sets that were obtained at the previous point 3.

Due to the fact that standard deviations of the $Ks$ values are in most cases unknown for meso- and macroscale hydrological model applications, this parameter should be assumed by the user. Note that this is the only tuning parameter needed for the procedure presented in this study. The standard deviations of $Ks$ values at field scale may vary between less than 50% and several hundred % and there seem to be no clear correlations to the texture classes of the analyzed soils, see e.g. Ciollaro and Romano (1995), Reynolds and Zebchuk (1996), Bosch, D. D. and West, L. T. (1998), Mohanty and Mousli (2000), Gupta et al. (2006) or Sobieraj et al. (2002). The range of the standard deviation that should be used is indirectly given by the minimum and



the maximum $Ks$ values that were obtained out of database $3_x$ (variant A). Assuming a specific standard deviation the 10% and 90% percentiles of the resulting $Ks$ distribution have still to be within the range of $Ks$ values given in database $3_x$ (variant A). If yes, the hydrological model is ready to start the simulation. If not, the regression function should either be restricted to the range of $Ks$ (this is recommended) or the standard deviation should be forced to a maximum value by the model. After

fulfilling this condition, the hydrological model is ready to start. A possibility to effectively process the VGP sets within the hydrological model is given in point 3 of the above list. We recommend to use at least 3 different VGP sets per soil to describe the spatially variability. However, more sets can be used likewise. It is easily possible to simulate the soil water movement for all VGP sets parallel in one simulation run of the hydrological model. The calculation time is therefore hardly affected. Note that vertical information about soil profiles, if available by the soil map, can be handled with the same procedure as described

so far. Hence, the spatial variability of soil hydraulic functions can either be described "horizontal" (if just texture classes without any vertical profile information is available) or "horizontal" + "vertical" (if soil profile information is available, too). These presented developments were implemented into the hydrological modelling system PANTA RHEI (Förster et al., 2012; Förster, 2013; Kreye et al., 2010, 2012; Kreye, 2015) and were used successfully in many practical applications and projects (e.g. Hölscher et al. (2014); Wurpts et al. (2014); Kreye (2015)). The structure of the soil model is shown in Fig. 9. Different

parametrizations of VGP are established by means of log normal distributions of $Ks$ . The resulting parametrizations are designated as domains and are parallel used at all spatial locations of the model area. The domains are solved simultaneously and with interaction to each other. The impact of the subgrid parameterization of the soil hydraulic functions are dominated by the variation of $Ks$ in wet periods and by the variation of VGP in dry periods. Furthermore, the parameterizations have a feedback on the reduction of evapotranspiration that can be related to the pressure head of the soil (Feddes et al., 1976).

The developed soil model is innovative regarding concept, interfaces, and parameterization. The model structure provides the required interfaces for calibrations made at runoff, satellite based soil moisture and groundwater level. Therefore, the demand for an automated optimisation procedure arises through the multi-variable examination of the system and its new complexity. A pioneering lexicographical strategy of optimisation was developed, using the model interfaces connected to modern data types (Gelleszun et al., 2015). Further results will be presented in a subsequent paper.

## 4  Conclusions

The objective of this study was to present a robust procedure to generate various sets of parameterizations of soil hydraulic functions for the description of soil heterogeneity on a subgrid scale. To achieve this, relations between $Ks$ values and van Genuchten's parameters of soil hydraulic functions were investigated. The VGP were obtained with the software ROSETTA. An universal function that is valid for the complete bandwidth of $Ks$ values could not be found. After concentrating on natural

texture classes, strong correlations were identified for all parameters. The results of the numerical study presented here confirm the findings of field studies (Li et al., 2007; Botros et al., 2009). The methodology presented in this study is applicable on a wide range of spatial scales and does not need input data from field studies.

Zhu and Mohanty (2002) tried to find effective parameters for van Genuchten's soil hydraulic functions within a numerical





study. They conclude that it is very difficult to define a single set of effective parameters that lead to suitable simulation results. In order to avoid effective parameters, the assumption of a parameterization of soil hydraulic functions in dependence of $Ks$, as presented in this study, is a promising alternative. Therefore, regression functions have to be set up a priori to the hydrological modelling. This is done in a much shorter time than the time needed for acquisition and preparation of other input data for a large scale hydrological model. Further, the procedure is robust in application and additional data (and costs) are not required. When using ROSETTA, a soil map of the modelling area is sufficient.

The procedure presented this study can be connected to the work of Wösten et al. (1985), Wösten et al. (1986), Carsel and Parrish (1988) and de Rooij et al. (2004). Wösten et al. (1985) and Wösten et al. (1986) successfully elaborated a procedure to regionalize soil hydraulic properties on the total model area by using measurement point data (for different soil profiles) and soil maps. However, in contrast to our work, they needed measurement data and the model area is very small (a few hundred hectares) compared to meso- and macroscale hydrological model areas with several thousand square kilometres. Besides texture data they used additional soil properties like bulk density or organic matter. The sophisticated methods for the consideration of subgrid variability presented by Carsel and Parrish (1988) and de Rooij et al. (2004) may be difficult to implement for hydrological modelling because of needed measurement data (again). However, for future work, it might be interesting to feed their methods with ROSETTA generated input data.

It is worth of discussion that the high correlations between ROSETTA generated $Ks$ and the other VGP are possibly artificially caused by ROSETTA itself. The neural network establishes relations between the percentage fractions of texture classes and the VGP each. This implicitly may lead to artificial correlations between the VGP. Looking at the citet field studies on the other hand, correlations seem to exist. However, assuming that ROSETTA actually boosts the correlations to a certain extent, - the message of this study is still the same. Compared to point measurements, ROSETTA is not always capable to perform a perfect fit, see e.g. Pandey et al. (2005), Li et al. (2007) or Ghorbani Dashtaki et al. (2010). Considering the huge sizes of model areas that are common for hydrological model applications however, ROSETTA is a good choice to generate parameters covering the complete area.





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

**Table 1.** Definitions of the used texture classes. The fractions of sand, silt and clay is processed out of the soil map for Lower Saxony (Edt.: J. Boess et al., 2004) and the German soil classification system (Sponagel, 2005).

| Abbreviation | Definition | Sand [%] | Silt [%] | Clay [%] |
|---|---|---|---|---|
| Lt | Clayey loam | 25 | 40 | 35 |
| Lu | Silty loam | 18.5 | 58 | 23.5 |
| Ls | Sandy loam | 44 | 35 | 21 |
| Ut | Clayey silt | 9 | 74 | 17 |
| Ul | Loamy silt | 27 | 58 | 15 |
| Us | Sandy silt | 32.5 | 65 | 2.5 |
| Sl | Loamy sand | 65 | 25 | 10 |
| Su | Silty sand | 63.5 | 32.5 | 4 |
| S | Sand | 85 | 10 | 5 |
| Ss | Pure sand | 92.5 | 5 | 2.5 |





**Table 2.** Obtained coefficients of determination ($R^2$), Spearman correlation ($r_{spear}$) and belonging p-value ($p$) as well as the sample size (Samples) for the regressions between the $Ks$ values and the soil hydraulic parameters for each texture class.

| Texture | Statistic | van Genuchten parameters | | | |
| --- | --- | --- | --- | --- | --- |
| | | $\Theta_R$ | $\Theta_S$ | $n$ | $\alpha$ |
| Lu | $R^2$ | 0.94 | 0.82 | 0.78 | 0.73 |
| | $r_{spear}$ | 0.97 | 0.91 | 0.86 | 0.88 |
| | $p$ | 0.00 | 0.00 | 0.00 | 0.00 |
| | Samples | 13829 | | | |
| Ls | $R^2$ | 0.88 | 0.90 | | |
| | $r_{spear}$ | 0.94 | 0.95 | | |
| | $p$ | 0.00 | 0.00 | | |
| | Samples | 50648 | | | |
| Ut | $R^2$ | 0.99 | | 0.93 | |
| | $r_{spear}$ | 1.00 | | 0.96 | |
| | $p$ | 0.00 | | 0.00 | |
| | Samples | 6822 | | | |
| Ul | $R^2$ | 0.98 | | 0.63 | |
| | $r_{spear}$ | 0.99 | | 0.79 | |
| | $p$ | 0.00 | | 0.00 | |
| | Samples | 12995 | | | |
| Us | $R^2$ | 0.99 | 0.78 | 0.56 | 0.96 |
| | $r_{spear}$ | 1.00 | 0.89 | 0.74 | 0.98 |
| | $p$ | 0.00 | 0.00 | 0.00 | 0.00 |
| | Samples | 3093 | | | |
| Sl | $R^2$ | 0.92 | 0.68 | 0.88 | 0.67 |
| | $r_{spear}$ | 0.95 | 0.83 | 0.96 | 0.80 |
| | $p$ | 0.00 | 0.00 | 0.00 | 0.00 |
| | Samples | 7202 | | | |
| Su | $R^2$ | 0.99 | 0.93 | 0.76 | 0.63 |
| | $r_{spear}$ | 0.99 | 0.96 | 0.92 | 0.78 |
| | $p$ | 0.00 | 0.00 | 0.00 | 0.00 |
| | Samples | 6364 | | | |
| S | $R^2$ | | | 1.00 | |
| | $r_{spear}$ | | | 1.00 | |
| | $p$ | | | 0.00 | |
| | Samples | 1455 | | | |
| Ss | $R^2$ | | | 0.98 | |
| | $r_{spear}$ | | | 0.99 | |
| | $p$ | | | 0.00 | |
| | Samples | 479 | | | |
| | Mean $R^2$ | 0.96 | 0.82 | 0.85 | 0.75 |
| | Mean $r_{spear}$ | 0.98 | 0.91 | 0.90 | 0.86 |





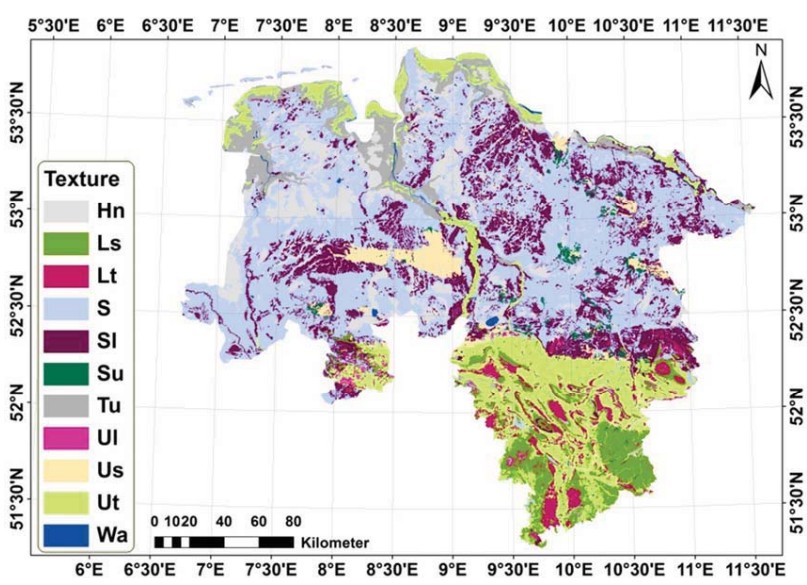

**Figure 1.** Soil map of Lower Saxony, Germany (Edt.: J. Boess et al., 2004). The abbreviations of the texture classes are explained in Table 1; in addition to that "Hn" stands for boggy soils and "Wa" stands for water bodies (lakes, rivers).





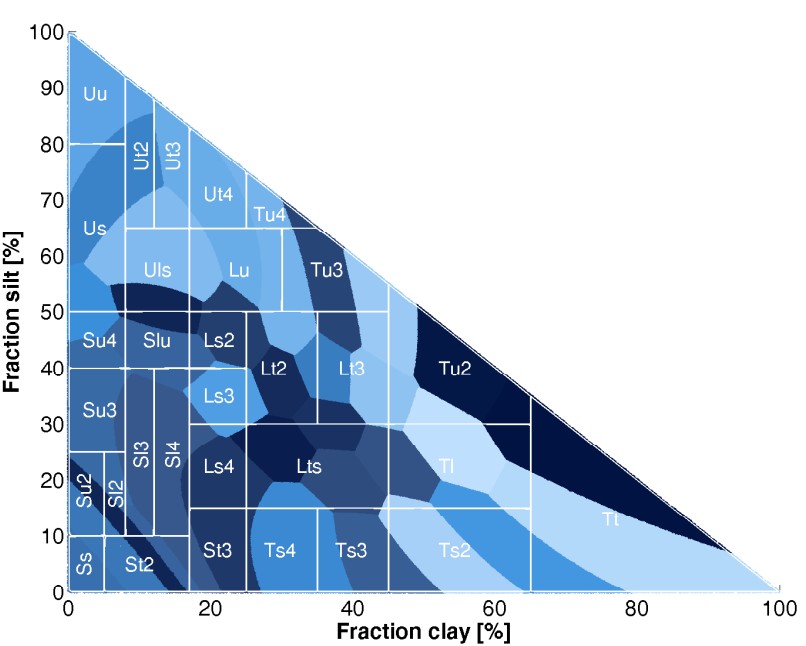

**Figure 2.** Subdivision of the soil texture by means of cluster analyses based on 31 classes (blue colored polygons). The classes were divided by similarity of their soil hydraulic parameters (cf. Twarakavi et al. (2010)). The subdivisions of the german soil classification system (cf. Sponagel (2005)) are overlayed with white lines.

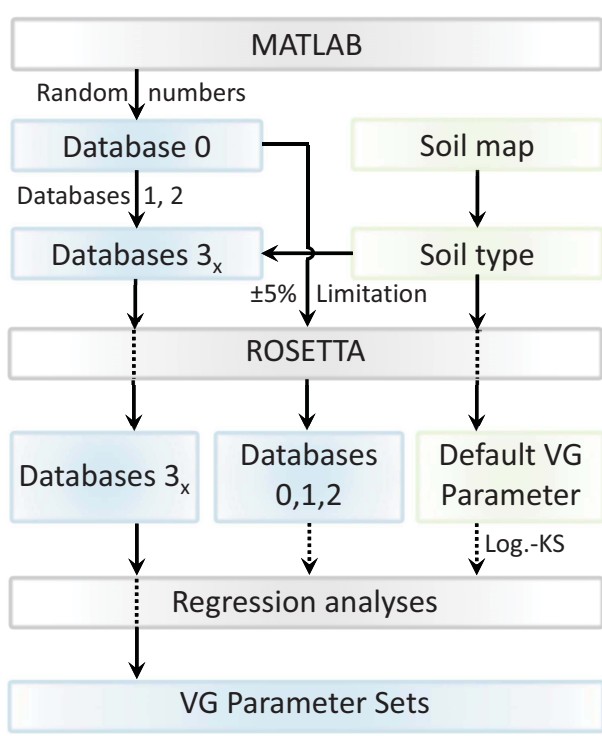

**Figure 3.** Procedure to obtain van Genuchten parameters and $Ks$ values based on soil map information. The Software ROSETTA is based on neural network analyses and generates van Genuchten parameters and $Ks$ values out of soil texture information.





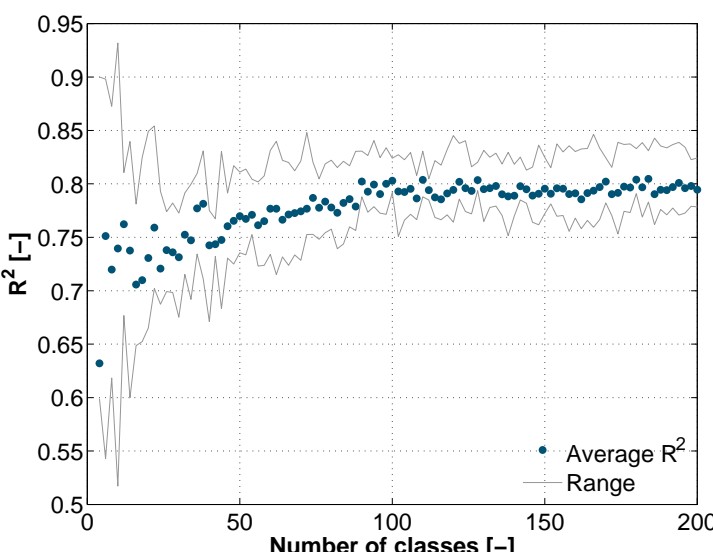

**Figure 4.** Average $R^2$ in dependency of the number of classes used for the subdivisions based on soil hydraulic properties by means of cluster analyses. The average $R^2$ is calculated out of the $R^2$ of all classes for each case. For this calculation, only classes with $R^2 > 0.5$ were considered. In addition to that, the range of $R^2$ is shown. The range yields out of the maximum and minimum $R^2$ of the individual classes.





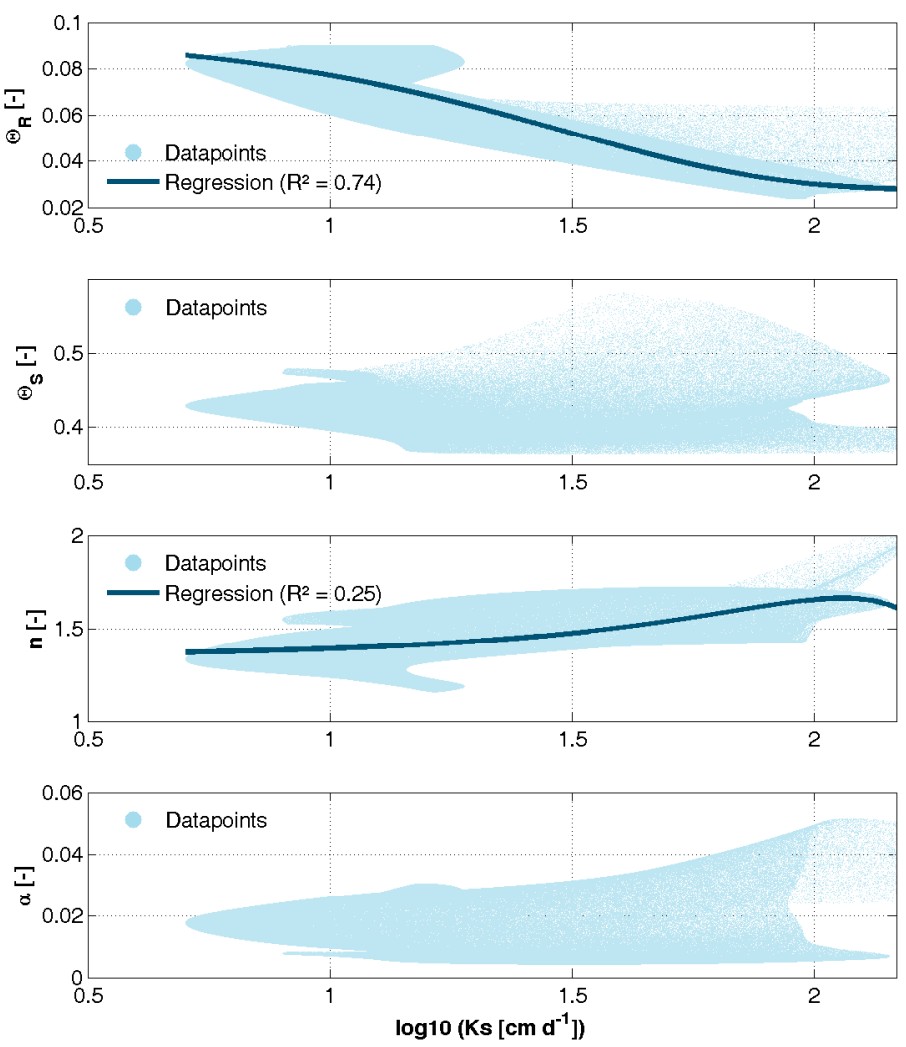

**Figure 5.** Scatterplots of the VGP in dependency of $Ks$. Visualized is database 1 ($\Theta_R - Ks$) and database 2 ($\Theta_S - Ks, n - Ks$ and $\alpha - Ks$). A regression function with an $R^2$ of 0.74 was fitted between $\Theta_R$ and $Ks$. Furthermore, a regression function with an $R^2$ of 0.25 was fitted between $n$ and $Ks$. $\Theta_S - Ks$ as well as $\alpha - Ks$ showed no correlation.





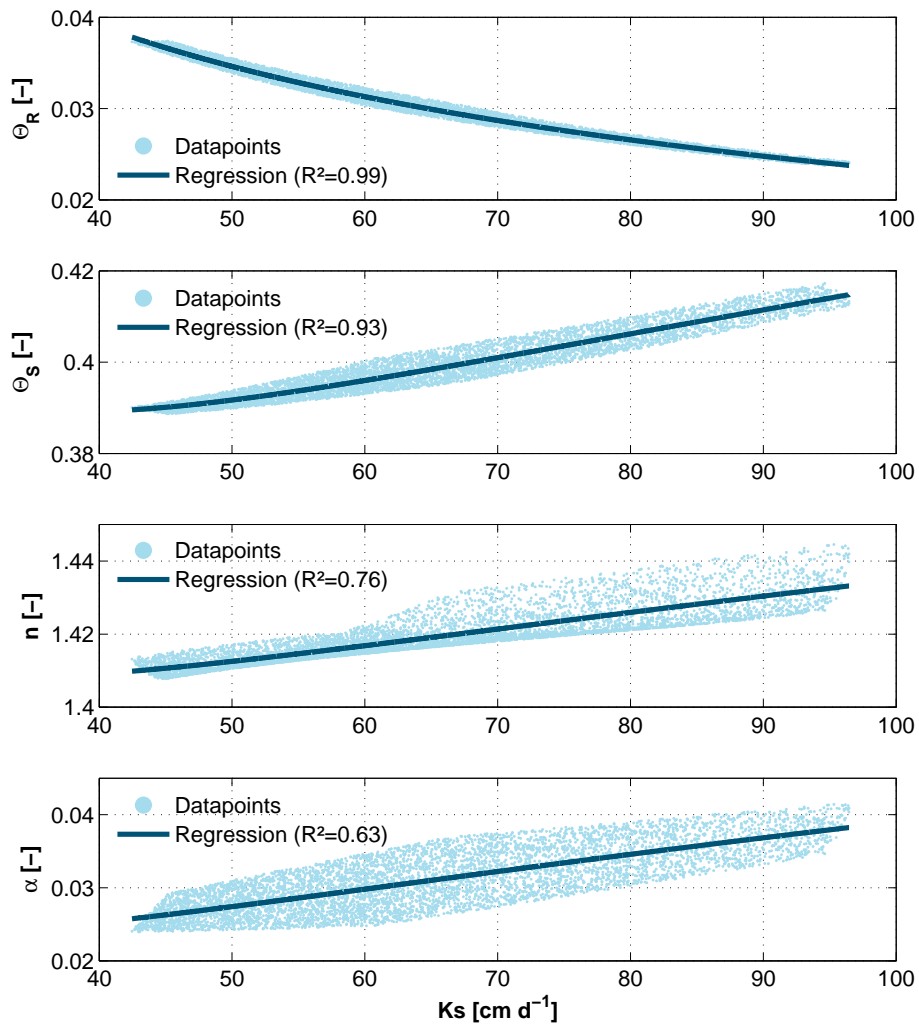

**Figure 6.** Scatterplots of the VGP in dependency of $Ks$ for the texture class Su out of database $3_x$ (variant A). Regression functions were fitted for all variants of VGP - $Ks$.





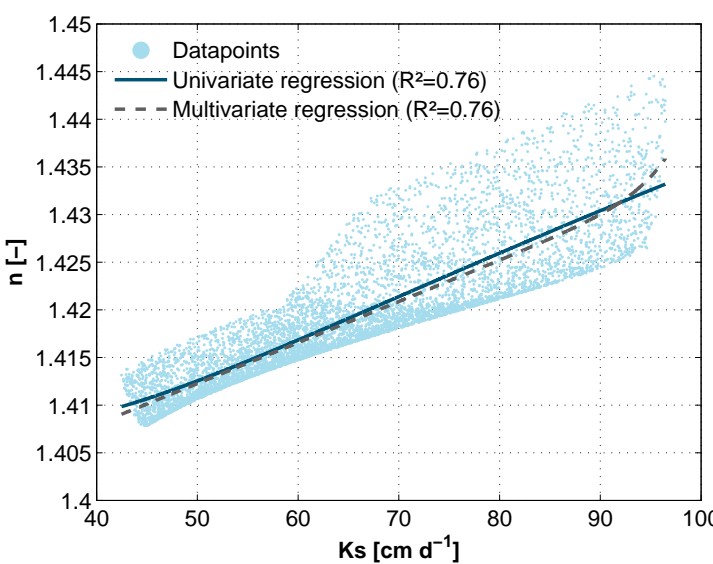

**Figure 7.** Scatterplot of $n$ in dependency of $Ks$ for the texture class Su out of database $3_x$ (variant A). To compare the univariate and multivariate regression, both functions are shown in the graph.





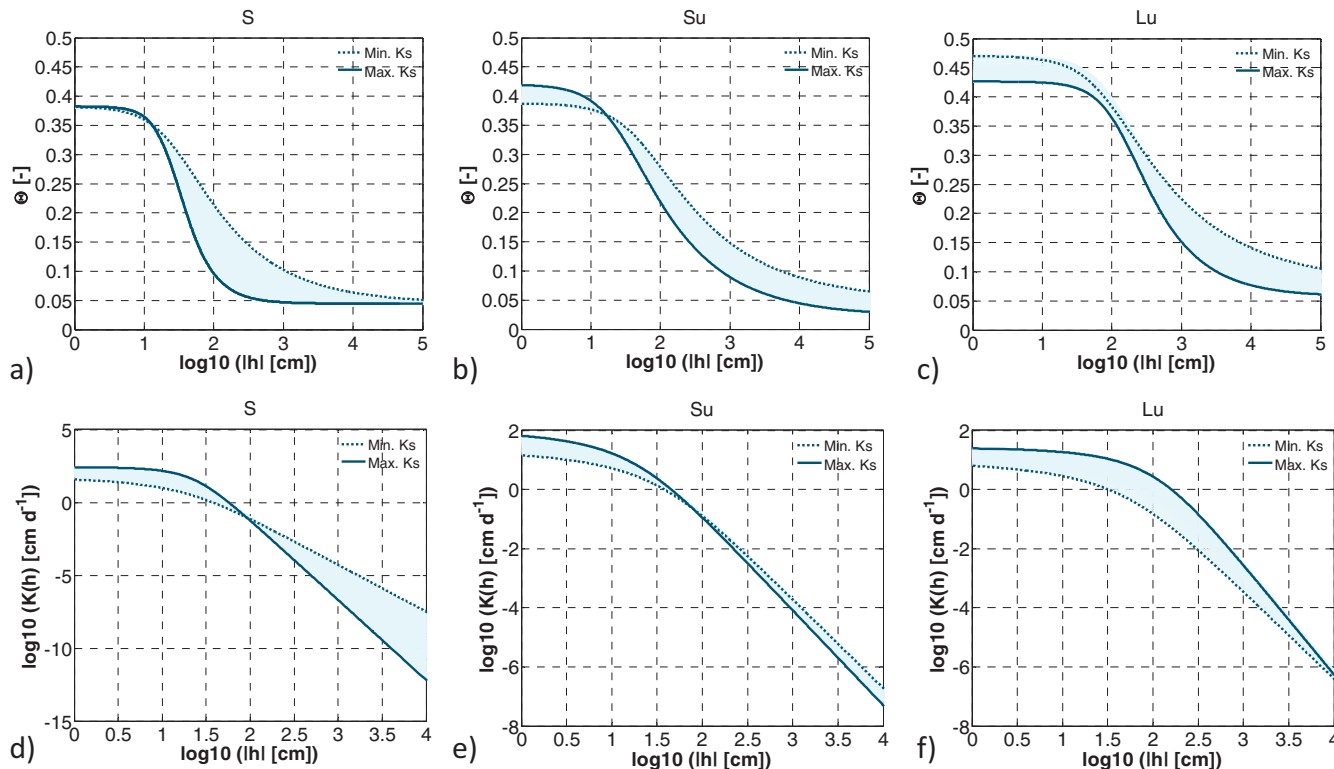

**Figure 8.** Impact on the pF and K(h) curves due to the univariate regression functions out of database $3_x$ (variant A). Minimum and maximum $Ks$ was given by ROSETTA. The VGP were changed in dependency of $Ks$ and the regression functions. a: pF curves for the texture class S. b and c: The same as shown in a, but for the texture classes Su and Lu. d: Unsaturated conductivity curves for the texture class S. e and f: The same as shown in d, but for the texture classes Su and Lu.





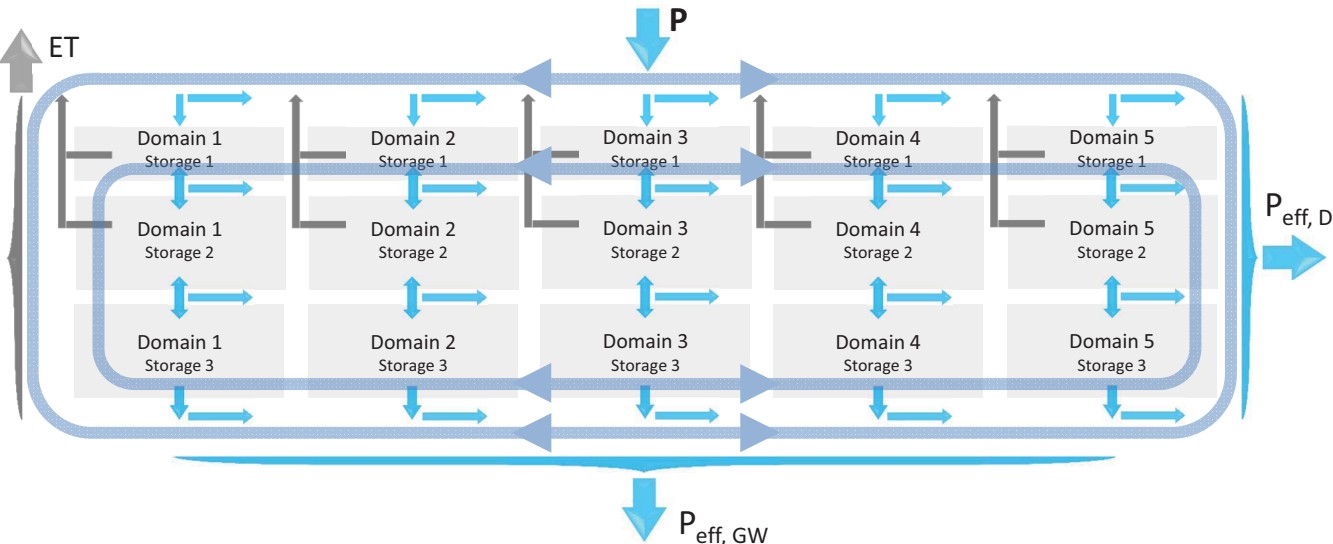

**Figure 9.** Application of different VGP parametrizations on the soil model of the hydrological modelling system PANTA RHEI. The different parametrizations (domains) are parallel used at all spatial locations. The domains are solved simultaneously and with interaction to each other. The main input is given by the spatial precipitation ($P$), which was reduced in advance by vegetational interception. Results of the soil model are the direct runoff ($P_{eff,D}$), the groundwater recharge ($P_{eff,GW}$), which leads to base flow in a long term view, and actual evapotranspiration ($ET$).