# Peer review of "Subgrid spatial variability of soil hydraulic functions for hydrological modelling"

_Hydrology and Earth System Sciences, 2016_

## Referee Comment (RC1) · Y. Gusev (Referee) · 20 Mar 2016

The objective of study presented in the paper is to develop a robust procedure to generate various sets of parameterizations of soil hydraulic functions for the description of soil heterogeneity on a subgrid scale. This goal is related to topics of the HESS journal, since it aims at improving the quality of hydrological models. Taking into account the subgrid effects (1) is necessary for increase of accuracy of calculations in the spatially distributed hydrological models, since output variables of the model are often associated with input variables and model parameters nonlinearly, (2) and also gives opens up a wider range of applications for solving related problems that require a description of the various processes at a smaller scale (assessment of intensity of erosion processes, the calculations of the pollutants transfer into the river system, accounting for hydrochemical processes in the catchment, etc.). When solving problems

of the describe the dynamics of water balance components in the distributed model the most important factors are the forcing data, i.e. hydrometeorological characteristics (primarily precipitation and incoming radiation (or the near surface air temperature correlating with it)) and the parameters of the soil cover. Namely to develop the methods of assessing the variability of the soil parameters within the framework of the computational grid of the hydrological model, this work is devoted. It aims to obtain data about the spatial variability of soil hydraulic parameters within the framework of the calculational cells of the catchment, when these data could be obtained indirectly, i.e. using only the information obtained from soil maps (reflecting the spatial distribution of soil classes) without experimental work on determination of the soil parameters. The main soil parameters that are considered in this work, were the parameters of the Mualem - van Genuchten model (VGP) (1980): namely the residual and saturated water content of the soil and the Ks, which is the saturated hydraulic conductivity. The sets of these parameters for different soil types, characterized by different textures (which is determined by the content of silt, sand and clay) were calculated with the use of the software ROSETTA (created in the Laboratory, Salinity, CA, USA; (Schaap et al., 2001)). In a next step, correlations between the Ks values and the VGP of the soil are analysed. As a result of the work, a procedure was developed to obtain the van Genuchten parameters (VGP) and KS values based on soil map information. Software Rosetta, based on neural network analysis, generates van Genuchten parameters and KS values from the soil texture information. In addition, the use of the Monte Carlo method gave the possibility to obtain for different classes of soil not only average values of the relevant parameters (the values of the Ks and VGP), as well as their variability. Obtaining this information is done in a much shorter time than the time required for the preparation and receipt of other input data for hydrological models of large spatial scales. In addition, the procedure is reliable in use and more data (and cost) is not required. When the code ROSETTA is used, it is enough only soil map. The developed technique can be used for any soil classification (e. g., UNSODA, USA; the German soil classification system (in this work), etc.). Presented in the work developments were implemented

into the hydrological modelling system PANTA RHEI (Förster et al., 2012; Förster, 2013; Kreye et al., 2010, 2012; Kreye, 2015) and were used successfully in many practical applications and projects (e.g. Hölscher et al. (2014); Wurpts et al. (2014); Kreye (2015)). It should be noted that although created technique was developed to account for soil heterogeneity, based on the soil parameters of the van Genuchten model, it can be applied when using other models, in which there are values of the $K_s$ and the residual and saturated water content of the soil. In conclusion, we believe that the paper makes a significant contribution to the development of hydrological models and may be published as submitted in the HESS journal.
* * *

---

## Referee Comment (RC2) · A. Zeyliger (Referee) · 22 Mar 2016

Review of article "Subgrid spatial variability of soil hydraulic functions for hydrological modeling" submitted by Phillip Kreye and Günter Meon (University of Braunschweig, Institute of Technology, Leichtweiss-Institute for Hydraulic Engineering and Water Resources, Dept. of Hydrology, Water Resources Management and Water Protection, , Beethovenstr. 51a, D-38106 Braunschweig, Germany), Email: (P.Kreye@tu-bs.de) to Hydrology and. Earth System Sciences

The article addresses an important question of effective parameterization of hydrological models by values of soil hydraulic property of empirical models. This study was focused on the analyzing of correlation between texture classes for German soil classification from one side and hydraulic soil property values of saturated soil hydraulic conductivities estimated for Mualem model (Mualem, 1976) in the version of application for van Genuchten pf-curve model (van Genuchten, 1980) and from other side of parameters of soil water retention curves fitted by the same van Genuchten model.

Fulfillment of this task was conducted in the base of the software ROSETTA (Schaap et al., 2001) with derived by neural network relationships within UNSODA soil data base (Leij et al., 1996; Nemes et al., 2001) linking USDA soil texture classes with values of soil hydraulic parameters.

Scientific significance

The research on this article is based on a some statistical analyses of correlations between parameters linked with ROSETTA within the UNSODA soil database.

In our opinion scientific significance of this research is limited by the use of the same database for analyzing relationships between saturated soil hydraulic conductivity and parameters of soil water retention curves where fitted. This limitation may be overcome by the use of another soil database like HYPRES to approve obtained results and may could provide a way for large verification of derived clusterization for any soil classes.

Scientific Quality

In our opinion a scientific quality is also limited by the use of parameters of specific empirical models describing shapes of both unsaturated soil hydraulic properties that are "not always valid".

First of all it is quite important for modeling infiltration to use adequate models for fitting and should be discussed more deeply according to some textural classes of soil and organic matter content. Thus is really important in many rainfall events for upper soil horizons with macropore structure controlling infiltration into soil profile which is not taken into account by selected Mualem-van Genuchten models.

Anatoly M. Zeyliger (Zeiliguer for old transliteration)
Russian State Agricultural University named after Timiriazev

azeiliguer@mail.ru

---

## Referee Comment (RC3) · Anonymous Referee #3 · 30 Mar 2016

Journal: HESS Title: Subgrid spatial variability of soil hydraulic functions for hydrological modelling Author(s): Phillip Kreye and Günter Meon MS No.: hess-2016-53 MS Type: Research article

General comments: Scientific significance: Good Scientific quality: Good Presentation quality: Good Recommendation: Return to author for major revisions.

The authors present research on the parameterization of physical based spatially distributed hydrological models that accounts for subgrid spatially variability. The procedure is well introduced and I agree that there is a need to advance spatial parametrization of hydrological models to represent heterogeneity of natural systems accordingly. The topic fits the scope of HESS, but, as detailed below, there are still a number of issues that require clarification before the manuscript can be accepted for publication.

[Figure]

Specific comments

1) The authors use ROSETTA to generate the databases of VGP sets based on trios of soil texture to build the regression models between Ks and the VGP. Overall, I see limited validity in this approach: (1) ROSETTA is a calibrated model which has effective parameters itself, as it is based on an imperfect model structure. This means that parameters found suitable for ROSETTA might not be very applicable in a distributed hydrological model. This was found by Koch et al. (2016) where parameters from a surrogate model (HYDRUS1D) were passed on to distributed models and it became clear that parameters are not easily interchangeable between models. (2) Along these lines it may be doubtful that the regression model between parameters of one model is transferable to another model. I would ask the authors to reflect on their assumption that the regression models found in ROSETTA are still valid in a more complex distributed hydrological model.

2) Section 3.3 nicely presents the workflow of the presented approach. However I would like to ask the authors to clarify how the VGP sets are incorporated in the hydrological model. Again, how can the authors support that the mean Ks value obtained from ROSETTA can be regarded as the mean Ks value for the more complex hydrological model, that may requires model dependent effective parameters (p.12,l.20). Instead a prior calibration of the hydrological model could be used to obtain suitable mean Ks values. How many sets of VGP sets should be used (p.12,l.24)? Also, the authors should give guidance how the subgrid spatial variability can be quantified after all VGP sets are executed (p.12,l.20)? The standard deviation of soil moisture at each cell?

3) Also I did not fully understand if the authors suggest having multiple model scenarios, where each scenario is based on a different Ks value drawn from the Ks distribution for each soil class? Or if they suggest to generate stochastic fields of Ks values that are applied in the distributed model?

4) In section 3.3 the authors address the problem of scale and that a pseudo accuracy

can be created if the model is operated at smaller scales than its input. Often model input comes at various scales and in fact hydrological processes take place at various scales as well. Here, the mHm model (Samaniego et al., 2010) provides a very flexible platform at account for differences in scale in the input data and parameters. The authors should mention modelling alternatives in their manuscript.

5) The authors mention that regression between Ks and the VGP could be artificially caused by ROSETTA. If this is the case, how do the authors support their suggested approach at all? What are the "real" regression models between Ks and other VGP and how wrong is ROSETTA? Again, this should be linked to the question if the same regression model can be assumed valid in a more complex hydrological model?

References

Koch, J., T. Cornelissen, Z. Fang, H. Bogena, B. Diekkrüger, S. Kollet, and S. Stisen (2016), Inter-comparison of three distributed hydrological models with respect to seasonal variability of soil moisture patterns at a small forested catchment, Journal of Hydrology, 533, 234-249.

Samaniego, L., R. Kumar, and S. Attinger (2010), Multiscale parameter regionalization of a grid-based hydrologic model at the mesoscale, Water Resources Research, 46.

---

## Author Comment (AC1) · 8 Apr 2016

We thank anonymous reviewer 3 for her/his detailed comments. This will help us to improve the manuscript.

**Specific comments**

> ROSETTA is a calibrated model which has effective parameters itself, as it is based on an imperfect model structure. This means that parameters found suitable for ROSETTA might not be very applicable in a distributed hydrological model. This was found by Koch et al. (2016) where parameters from a surrogate model

*(HYDRUS1D) were passed on to distributed models and it became clear that parameters are not easily interchangeable between models.*
*Along these lines it may be doubtful that the regression model between parameters of one model is transferable to another model.*
*I would ask the authors to reflect on their assumption that the regression models found in ROSETTA are still valid in a more complex distributed hydrological model.*

First, we would like to add some information regarding ROSETTA. This software is based on neural network analyses and servs as pedotransfer function for the estimation of van Genuchten water retention parameters (VGP) and the saturated hydraulic conductivity (Ks). Data with different level of detail can be used as input, starting with texture classes and going up to more detailed (experimentally determined) information (Schaap et al., 2001). However, ROSETTA doesn't fit VGPs and Ks by means of measured time series of e.g. soil moisture or pressure head. Hence, we prefer to define ROSETTA as "pedotransfer function" rather than using the term "model". Koch et al. (2016) used the model HYDRUS 1D to fit VGPs (thank you for this reference). This was done by means of continuously measured time series of soil moisture at different locations and depths. HYDRUS also incorporates a ROSETTA interface, but here inverse modelling was used to fit VGP. I totally agree with you, that it could be troublesome to transfer the VGP, which were determined by this manner, from HYDRUS to more complex hydrological models (but this isn't what we did). To parametrize their model, Koch et al. (2016) homogeneously used the same VGP at every spatial location for hydrological modelling. In a second (heterogeneous) scenario they used spatially differentiated porosity (saturated water content), but all other VGP are still homogenously distributed. Hence, they nicely conclude that "*future work must focus on other possibilities to further distribute the remaining VGM parameters*". One possibility to achieve this on the mesoscale is what we introduce in

our study. Summary: We use ROSETTA as pedotransfer function to estimate VGP.

> *Section 3.3 nicely presents the workflow of the presented approach.
> However I would like to ask the authors to clarify how the VGP sets
> are incorporated in the hydrological model. Again, how can the
> authors support that the mean Ks value obtained from ROSETTA
> can be regarded as the mean Ks value for the more complex
> hydrological model, that may requires model dependent effective
> parameters (p.12,l.20). Instead a prior calibration of the hydrological
> model could be used to obtain suitable mean Ks values.*

We feed ROSETTA with texture information based on soil maps (in our case, the soil map of Lower Saxony, 1:50.000). Therefore, the Ks values estimated by ROSETTA are effective that are values valid for the spatial resolution of the soil map. The simulations of soil water dynamics inside the hydrological model operate on the same spatial resolution as the soil map, because the spatial distribution of our hydrological model (PANTA RHEI) is based on polygons. To establish subgrid variability, we create distribution functions of Ks and VGP as described in the manuscript. But (and this is an important fact), we don't change the effective VGP/Ks set in order to calibrate the hydrological model.

> *How many sets of VGP sets should be used (p.12,l.24)?*

The number of sets is up to the user. At least three sets should be used. In our manuscript we recommend five sets by using the 10%, 30%, 50%, 70% and 90% percentile of the Ks distribution function. Of course, more sets are possible.

> *Also, the authors should give guidance how the subgrid spatial variability can be quantified after all VGP sets are executed (p.12,l.20)? The standard deviation of soil moisture at each cell?*

Yes, a possibility to account for subgrid variability is to analyse the standard deviation of soil moisture as a function of the number of applied VGP sets. Further, the spatial soil moisture patterns could be compared in dependence of the number of applied VGP sets, similar to Samaniego et al. (2010) (thank you for this reference). We compared breakthrough curves (1D) with different numbers of VGP sets and with different standard deviations of the Ks distribution functions. We also compared spatially distributed simulation results of the hydrological model for soil moisture with remotely sensed satellite data, but this goes beyond of this study. We are working on a pursuing manuscript focusing on the hydrological model and its calibration.

> *Also I did not fully understand if the authors suggest having multiple model scenarios, where each scenario is based on a different Ks value drawn from the Ks distribution for each soil class? Or if they suggest to generate stochastic fields of Ks values that are applied in the distributed model?*

You are right, we have to be more precise. After the different sets of VGP (e.g. 5) are derived, we use all of them to parameterize the soil model, which is incorporated in the hydrological model (PANTA RHEI). We assume, that one effective set of VGP cannot express subgrid variability (as described in the manuscript). Secondly, we assume, that many different sets of VPG are able to do so. That's why the soil model is parameterized many times, whereby the structure and equations were not changed. These different models (domains) operate simultaneously and at the same spatial location and are connected to each other. Please take a look to the attached figure. At

every spatial location (the resolution is determined by the soil map) we have different effective VGP and for every spatial location we parameterize 5 different VGP sets. The attached figure shows the idea at an abstract level, in fact our model is polygon based (and not grid based). Summary: we don't have multiple model scenarios. It is one model with multiple parameterizations.

> *In section 3.3 the authors address the problem of scale and that a pseudo accuracy can be created if the model is operated at smaller scales than its input. Often model input comes at various scales and in fact hydrological processes take place at various scales as well. Here, the mHm model (Samaniego et al., 2010) provides a very flexible platform at account for differences in scale in the input data and parameters. The authors should mention modelling alternatives in their manuscript.*

We didn't want to focus too much on the hydrological model. However, we agree to add more information here. Thank you again for the reference, we will pick this up in our manuscript. A big difference between our hydrological model PANTA RHEI compared to many other models is the number of model parameters that are used for calibration. We work with catchment based model parameters, which have different effects on the sub-catchment scale controlled by physiographic characteristics. This leads to (only) 6-8 model parameters in total to calibrate the model for an area of a few hundred square kilometres.

> *The authors mention that regression between Ks and the VGP could be artiflcially caused by ROSETTA. If this is the case, how do the authors support their suggested approach at all? What are the "real" regression models between Ks and other VGP and how wrong*

[Figure]

*is ROSETTA? Again, this should be linked to the question if the same regression model can be assumed valid in a more complex hydrological model?*

Results of ROSETTA are estimations and anyhow effective values. These effective values are "never correct" if compared to experimentally derived ("real") values. However, the advantages of ROSETTA are scale equality and that no experimental measurements are necassary. For that reason, these parameters are suitable for hydrological modelling. But, if we use one VGP set it may not be possible to describe all conditions of soil water in a plausible way. For instance, soil water dynamics could be well approximated for wet situations, but provide inadequate simulations for dry situations (this was also a problem of the simulations performed in Koch et al. (2016)). Hence, we use more than on set of VGP.

Connections between (experimentally derived) Ks and VGP are found in many studies, as described in our manuscript. Using ROSETTA we find quite strong connections. That's why we discuss at which proportion this could be enhanced by the artificial network. However, we think, that even if ROSETTA boost the connections between Ks and VGP it is admissible to generate distribution functions based on these connections as our focus is finding different "possibilities" to describe soil hydraulic behaviour within a certain framework.

**References**

Koch, J., Cornelissen, T., Fang, Z., Bogena, H., Diekkrüger, B., Kollet, S., and Stisen, S.: Inter-comparison of three distributed hydrological models with respect to seasonal variability of soil moisture patterns at a small forested catchment, Journal of Hydrology, 533, 234–249, 2016.

Samaniego, L., Kumar, R., and Attinger, S.: Multiscale parameter regionalization of a grid-based hydrologic model at the mesoscale, Water Resources Research, 46, n/a, 2010.

Schaap, M. G., van Leij, J. F., and van Genuchten, M. T.: ROSETTA: a computer program for estimating soil hydraulic parameters with hierarchical pedotransfer functions, Journal of Hydrology, 2001, 163–176, 2001.

[Figure]

**Fig. 1.**

---

## Short Comment (SC1) · 11 Apr 2016

The subject of the paper is relevant as hydrological properties of soils are of high importance in hydrological modeling. Scale problems are often ignored when large scale hydrological models are setup, partly because no soil-map preprocessing is made before the model setup and partly because of a lack of "easy access" methods for overcoming scale problems.

The work is relatively well presented in a clear language. The introduction is focused and gives a good overview. The text can in some sections be hard to follow for a non soil-scientist as my self (hydrological modeler) because of the large number of symbols and abbreviations.

Generally the figure captions needs to explain the symbols used in the figures, to com-

ply with the "being able to stand alone" criteria.

How is the Ks and other values included in the hydrological model? And how does the model handle the parameter-variation within each soil class (if I understand you right)?, Please give an example of the difference the use of different parameter settings makes to the model outcome. Please state what the hydrological model is evaluated against.

I share the concerns of reviewer 3, comment 5.

Few specific comment: page 8 line 28, "than instead of "as" 9/6 How can the multivariate method give a worse fit than the linear? 11/20 respectively instead of each

---

## Author Comment (AC2) · 11 Apr 2016

We thank reviewer 2, Anatoly M. Zeyliger, for his detailed comments. This will help us to improve the manuscript.

**Specific comments**

> *The research on this article is based on a some statistical analyses of correlations between parameters linked with ROSETTA within the UNSODA soil database.*
> *In our opinion scientific significance of this research is limited by the use of the same database for analyzing relationships between*

*saturated soil hydraulic conductivity and parameters of soil water retention curves where fitted. This limitation may be overcome by the use of another soil database like HYPRES to approve obtained results and may could provide a way for large verification of derived clusterization for any soil classes.*

As a pedotransfer function, ROSETTA works independent from its databases, which were used for its calibration. Besides UNSODA two other soil databases were used to calibrate the neural network of ROSETTA (Schaap et al., 1998). Of course you are right that our results are valid within the framework of ROSETTA only. However, parameterization of soil hydraulic functions on the hydrological meso- and macroscale are always based on one database or pedotransfer function. To have more variable van Genuchten parameter (VGP) sets, we use the method introduced in our manuscript. An inclusion of more databases (like HYPRES) is a good idea, but this demands for different methods (in our point of view).

*In our opinion a scientific quality is also limited by the use of parameters of specific empirical models describing shapes of both unsaturated soil hydraulic properties that are "not always valid".*

We are not sure, if we understand your statement correctly. We don't use the empirical regression functions or their parameters to describe the shapes of pF and conductivity curves directly. The regression functions are only used to vary VGP. In case of low or no correlations between the saturated hydraulic conductivity and a VGP, we don't change this VGP.

*First of all it is quite important for modeling infiltration to use*

*adequate models for fitting and should be discussed more deeply according to some textural classes of soil and organic matter content. Thus is really important in many rainfall events for upper soil horizons with macropore structure controlling infiltration into soil profile which is not taken into account by selected Mualem-van Genuchten models.*

Yes, you are right that Mualem-van Genuchten is valid for matrix flow. However, the calculation module of infiltration and percolation within the hydrological model PANTA RHEI uses different pathways to account for preferential flow (Kreye, 2015). In addition to that, using different sets of VGP at the same spatial location has a similar effect: We don't have a "homogeneous" soil matrix. If e.g. soil moisture conditions are high, the VGP set with high saturated conductivity becomes dominant. In dry conditions, it could be the other way round.

**References**

Schaap, M. G., Leij, F. J., and van Genuchten, Martinus Th.: Neural Network Analysis for Hierarchical Prediction of Soil Hydraulic Properties, Soil Sci. Soc. Am. J., 62, 847–855, 1998.

Kreye, P.: Mesoskalige Bodenwasserhaushaltsmodellierung mit Nutzung von Grundwassermessungen und satellitenbasierten Bodenfeuchtedaten, Braunschweig, 2015.

---

## Author Comment (AC3) · 12 Apr 2016

We thank Dr. Hans Thodsen for his comments. This will help us to improve the manuscript.

**Specific comments**

> *The text can in some sections be hard to follow for a non soil-scientist as my self (hydrological modeler) because of the large number of symbols and abbreviations.*

We could add a short appendix with a list of abbreviations, if the editor agrees.

[Figure]

*Generally the figure captions needs to explain the symbols used in the figures, to comply with the "being able to stand alone" criteria.*

We totally agree and are going to improve the figure captions.

*How is the Ks and other values included in the hydrological model? And how does the model handle the parameter-variation within each soil class (if I understand you right)? Please give an example of the difference the use of different parameter settings makes to the model outcome. Please state what the hydrological model is evaluated against.*

In the following, we partly use similar explanations as in our answer to reviewer 3, which we uploaded just shortly before you made your comment.
We use all derived sets of van Genuchten parameters (VGP) and Ks to parameterize the soil hydraulic functions of the soil model, which is incorporated in the hydrological model (PANTA RHEI). Hence, the soil model is parameterized many times with different VGP sets for the same location; the structure and equations were not changed. These "different" models (domains) operate simultaneously and are connected to each other. Summary: We have one model with multiple parameterizations (please take a look at the attached figure in our answer to reviewer 3).
We compared breakthrough curves (1D) with different numbers of VGP sets and with different standard deviations of the Ks distribution functions. Soil moisture patterns could also be compared in dependence of the number of applied VGP sets. The model is evaluated against three types of observations/data: discharge, spatial distributed soil moisture (satellite data, ERS1/2-ESCAT, MetOp-ASCAT, ENVISAT-ASAR) and

groundwater level. The satellite soil moisture data accounts for the upper few cm of the soil surface. The groundwater level data accounts for the lower boundary of the soil model. Individual objective functions were used and connected to a stepwise Downhill-Simplex to calibrate the model parameters. To achieve this, a lexicographical strategy was developed, where different objectives can be defined as an order of preference (Gelleszun et al., 2015). At the moment, we are working on a pursuing manuscript focusing on the hydrological model and its calibration.

*9/6 How can the multivariate method give a worse fit than the linear?*

The average $R^2$ are nearly the same (rounded numbers). We think that both methods achieve the same performance and the small difference in $R^2$ can be explained by the optimization algorithm (we used Levenberg-Marquardt, which is a local algorithm).

**References**

Gelleszun, M., Kreye, P., and Meon, G.: Lexikografische Kalibrierungsstrategie für eine effiziente Parameterschätzung in hochaufgelösten Niederschlag-Abfluss-Modellen, Hydrologie und Wasserbewirtschaftung, 59, 84–95, 2015